# The Links Between Community-Based Financial Inclusion and Household Food Availability: Evidence from Mozambique

**DOI:** 10.3390/foods14020212

**Published:** 2025-01-12

**Authors:** Aweke Tadesse, Kenan Li, Jesse Helton, Jin Huang, David Ansong

**Affiliations:** 1School of Social Work, Saint Louis University, St. Louis, MO 63103, USA; jesse.helton@slu.edu (J.H.); jin.huang@slu.edu (J.H.); 2Department of Epidemiology and Biostatistics, Saint Louis University, St. Louis, MO 63103, USA; 3School of Social Work, The University of North Carolina at Chapel Hill, Chapel Hill, NC 27516, USA; ansong@email.unc.edu

**Keywords:** financial inclusion, village saving and loan group, food insecurity, household hunger, household assets

## Abstract

Financial inclusion can boost wealth, health, and quality of life. However, few studies have examined how women’s participation in community-based financial inclusion opportunities, such as village saving and loan groups (VSLGs), relates to household food security. Using program data from central Mozambique, this study examined whether low-income women’s participation in VSLGs directly increases household food availability, as well as indirectly through increased asset ownership. Employing a post-test-only comparison group quasi-experimental design, the study sampled 205 female VSLG participants and non-participants from three sub-villages in Mozambique’s Sofala province. Structural equation modeling (SEM) results indicated that low-income women’s participation in VSLGs is directly associated with a reduction in household hunger score (β = −0.21, *p* < 0.01), as well as indirectly associated through the mediating role of household assets ([Sobel indirect effect] = −0.06, *p* = 0.05). The VSLG participants showed a significant increase in household asset ownership compared to non-VSLG participants (β = 0.15, *p* < 0.05). Further, increased asset ownership significantly correlated with a lower probability of household hunger (β = −0.30, *p* < 0.01). The results suggest that community-based financial inclusion approaches could improve the availability of food through asset building among Mozambique’s low-income women. The study offers a potential strategy for policymakers and development experts to utilize community approaches to financial inclusion to improve rural and low-income women’s livelihoods.

## 1. Introduction

Economic development is crucial in driving human well-being and institutional change, with significant implications for various well-being domains such as health, wealth, education, and food security [1,2]. Food insecurity represents a global health concern, resulting in an estimated 300,000 deaths each year, and it was estimated that the prevalence of food insecurity would reach 17 million by 2021, with 50% of child mortality in the Africa attributed to inadequate access to quality food [3]. After the COVID pandemic, reports from the Food and Agriculture Organization (FAO) and World Health Organization (WHO) estimated that annually 9 million people die from hunger and hunger-related issues globally, including deaths associated with health complications [4,5]. Food insecurity trends have worsened over the past two decades, with the number of malnourished children alone increasing from 5.5 million to 30 million. Tragically, this inadequate food consumption has led to the death of 3.5 million children under the age of five in Sub-Saharan Africa (SSA) [6].

In SSA, various community-driven financial capability strategies were introduced for tackling rural poverty and other related socioeconomic limitations. For example, the microfinance cooperatives that provided small loans and an opportunity for savings enabled farmers to gain inputs towards farming equipment, seeds, and fertilizers, which in turn have a direct positive impact in production and productivity and in improving the availability of food in the household and in increasing resilience [7]. Similarly, other informal financial inclusion approaches in Africa, such as micro-loan and savings (MLS), rotating savings and credit associations (ROSCAs), and village savings and loan groups (VSLAs), were implemented for fostering small-scale income generation activities that allowed rural farming households and others to purchase livestock, withstand seasonal shocks, and improve their food security [8].

In Mozambique, the setting for our study, disadvantaged populations such as single women and female caregivers of orphans and vulnerable children (OVC) face severe crises and economic hardships, exacerbating a range of complex well-being challenges [9]. Mozambique’s global rankings in maternal mortality (20th, with 489 deaths per 100,000 live births) and under-five mortality (13th, with 67.9 deaths per 1000 live births) in 2016 further underscore the challenges faced [10]. Additionally, the 2018 Human Development Index positioned Mozambique at 180 out of 189 countries with low quality of life, highlighting the need for significant improvements [10,11]. Disturbingly, reports indicate that 43% of children under the age of five in the country suffer from chronic malnutrition (8% acute malnutrition), and over 65% of the population lacks access to two daily balanced meals, potable water, basic sanitation, and adequate housing [12]. Recognizing the significance of addressing financial inclusion for marginalized populations in low-income regions, the 2011 Maya Declaration has played a crucial role in promoting inclusive policies, particularly for the poor [13].

Theoretical and empirical studies have consistently shown that financial inclusion (FI) is a critical determinant of poverty alleviation and its multifaceted effects, including food security. FI refers to the process of providing accessible and affordable opportunities for financial systems and services [14]. However, globally, FI is not fully utilized, particularly in low-income countries (LICs), where approximately 2.7 billion people are still excluded [15]. The gap is even worse in Africa, with less than half (23%) of the global FI estimate being achieved [14]. Access to financial services offers opportunities to overcome obstacles to both individual (micro) and collective (macro) growth, enabling individuals to secure a stable living standard. Despite the macro and micro benefits of FI in improving living standards, Africa has the lowest level of FI [16], with a significant gender disparity that hampers women’s economic autonomy [17].

Poverty and economic hardships are common barriers to sustainable development and healthy living worldwide [13]. In low-income communities, formal and informal financial inclusion can address life-threatening shocks that hinder healthy living, depending on demographic and socioeconomic characteristics and accessibility to financial services [18,19,20]. This study proposes FI as a mechanism to tackle the issue of food availability in households, encompassing factors such as affordability and access. Specifically, it examines the association between a community-based financial inclusion (IFI) opportunity (through participation in village savings and loan groups (VSLGs)) and household food security (HFS) among low-income women in Mozambique and accounts for the intermediary role of household asset ownership.

### 1.1. Financial Inclusion Targeting the Burden of Food Insecurity

Globally, financial inclusion has been on the rise [21]. The global 2021 Findex 10-year assessment (2011 to 2021) indicated that financial account ownership increased by 49%, from 51% to 76% [22]. The report also noted an 8% increase in Africa, with account ownership rising from 63% to 71%, and 50% of adults in SSA now use a mobile account system. Notwithstanding the remarkable increase in formal bank accounts, this figure is not evenly distributed between genders: the percentage of account holders is 8% lower for women. Also, a financial inclusion assessment from 20 African countries revealed significant limitations in the benefits derived from existing formal financial services. This study reported that a low percentage of the population has access to loans, debit cards, and credit. Notably, Mozambique was among the countries with the lowest use of formal financial services for loans (37%), debit cards (6%), and credit (4%).

Notwithstanding the inequitable access challenges, financial inclusion can potentially improve well-being. The World Bank has championed financial inclusion as a global strategy to reduce poverty and economic disparity [23,24,25,26]. Financial inclusion facilitates access to services like loans, payments, insurance, and savings at the individual and household levels. These services can further stimulate wealth and asset accumulation, informing investment decisions, increasing food security, and enhancing purchasing power [27,28,29]. However, poverty is highly concentrated among rural communities with limited access to formal financial institutions, impeding inclusive development. This dynamic exacerbates food insecurity, influencing consumption, nutrition, and the physical availability of food [30]. Research from 88 low-income countries suggests that access to financial inclusion enhances the likelihood of savings, credit utilization, and risk-taking for investments, potentially bolstering resilience and impacting the quality of life in physical, social, mental, economic, and psychological well-being [31,32]. However, rural communities in Africa continue to grapple with poverty and its associated impacts, lagging in financial inclusion. This study, therefore, seeks to investigate how access to a financial inclusion opportunity could address the persistent issues of food insecurity and recurring hunger.

### 1.2. Financial Inclusion and Gender Disparities

For this study, financial inclusion denotes individuals’ access to beneficial formal financial products and services (including credit, savings, and payment services) [33,34]. Across the world, but more so in developing economies and rural contexts, there remain persistent unequal opportunities between women and men in access to formal financial services, stemming mostly from non-biological characteristics such as beliefs, cultural norms, and societal values [35]. For instance, the percentage of adult women with formal financial accounts (approximately 20%) is 15% lower than that of adult men (approximately above 35%). This discrepancy in financial inclusion, or financial exclusion, limits the ability of low-income individuals and households to make productive investment decisions, impacting household food security and availability [36,37,38].

Studies from Sub-Saharan Africa have shown that financial inclusion and empowerment interventions increased the likelihood of food security among low-income Ghanaian women [39]. There is substantial evidence supporting the correlation between women’s financial inclusion and enhanced financial outcomes, improved interpersonal relationships, and a decrease in domestic and intimate partner violence (IPV) [33,34,40].

Due to complex cultural, social, and economic disparities, women’s financial inclusion in Africa is generally lower (20%) compared to men (27%). However, women often manage earnings, savings, and large resources for household use more effectively, covering consumption, health, education, and other family needs [41]. Reports suggest that compared to men (1 in 4), fewer women (1 in 5) have access to formal financial institutions and hold an account [17]. Financial inclusion and economic participation are vital to promoting gender equity and supporting women’s and caregivers’ economic rights to achieve a suitable standard of living [42]. Thus, it is critical to test and scale interventions and programs that foster everyone’s financial management skills and access to and use of beneficial financial policies and services, expanding everyone’s asset-building and economic development opportunities.

### 1.3. Village Savings and Loan Groups

Financial inclusion has both macro- and micro-level outcomes that contribute to collective and subjective well-being and living standards through financial and non-financial means. At the household level, financial inclusion provides opportunities for saving, credit, and investment, enhancing coping skills during times of uncertainty and shocks [43,44]. At the community level, village savings and loan groups (VSLGs) play a pivotal role in financial inclusion. The VSLGs are community-managed informal microfinance cooperatives that have proven to be adaptable in rural African contexts, particularly in villages where 15 to 35 individuals come together to form a group [45]. Unlike traditional banks, which often require extensive documentation, such as proof of residency, income records, credit scores, and formal identification, VSLGs primarily rely on community trust and personal relationships. In a VSLG, participants do not need to present financial history or meet rigid administrative requirements to become a member. The informal nature of these groups makes entry accessible to people who may not have the necessary documentation for formal banking [46].

VSLGs in Sub-Saharan Africa (SSA) serve as an informal financial inclusion mechanism that often provides low-income populations with opportunities for small-scale savings, loans, and profit-sharing, while improving financial decision-making skills and behaviors, including risk-taking in investments [6]. VSLGs serve as an alternative for the low-income community to bridge the economic gap, improve livelihoods, and enhance the quality of life by promoting changes in income, assets, investments, and food security, ultimately increasing productivity and purchasing potential [6,47,48].

Despite the potential contributions of VSLGs in addressing food insecurity, there is a scarcity of information linking the financial and interpersonal behavioral outcomes of VSLGs with food availability among low-income households. This study aims to fill this gap in knowledge by using data from Mozambique to examine the association between VSLG participation and food availability.

### 1.4. Household Assets

There is a clear link between food security and disparities in asset ownership, wealth, and poverty, including a significant association between lower household wealth and increased food insecurity. Household assets are conceptualized as financial resources and commodities that positively impact household members. Studies have suggested that, beyond financial gain, asset-building interventions significantly improve household and individual health conditions and behaviors [49,50].

Ownership of certain assets can be a source of income and sustainable savings [51,52], reducing financial hardship and poverty outcomes. In Sub-Saharan Africa (SSA), typical material assets include cars, televisions, farmland, domestic animals, houses, refrigerators, washing machines, computers, dishwashers, bicycles, mobile phones, fixed telephones, and other kitchen items. However, the range and quality of these assets may vary between urban and rural areas and among low-income groups.

### 1.5. Current Gaps in Modelling Approach

Evidence from both low-income and high-income nations suggests that outcomes of FI interventions, such as promoting savings and social skills, break the cyclical risk of poverty and increase the likelihood of improving individuals’ living conditions [53]. Over time, savings and accumulated assets, whether at the individual or community level, facilitate micro-investments that can have a significant impact on personal and household income, assets, and overall wealth, ultimately affecting food consumption and food security [54].

In addition to economic theories, social causation theory has been employed to explain the mechanisms that predict non-financial outcomes, including food security [55,56,57,58]. According to this theory, factors such as group membership, shared norms, and social networks could influence individuals’ financial decisions, access to resources, and overall economic success [59]. Limited access to formal financial services, particularly in low-income rural areas, presents a barrier that prevents individuals from accessing opportunities that could potentially help them escape poverty and improve their living conditions. Participation in VSLGs provides an avenue for accumulating savings and micro-investments, which can positively impact individual income, assets, and overall wealth. These outcomes are associated with improved subjective well-being and household quality of life, including enhanced purchasing power and food security [60].

Therefore, as illustrated in Figure 1, this study hypothesizes that compared to non-VSLG members, participants of VSLG have a higher probability of food availability (as indicated by a lower hunger score). Additionally, it is hypothesized that participation in VSLG increases asset ownership, which, in turn, increases food availability. Despite the potential positive impact of VSLG participation on financial and non-financial outcomes, there is a dearth of studies investigating this specific strategy and its effects on food security, particularly concerning food availability among low-income women in Mozambique. This study aims to fill this knowledge gap by examining the association between financial VSLG participation and food availability among low-income adult women in Mozambique, considering the mediating role of household assets.

## 2. Data and Methods

### 2.1. Data Collection and Variables

The data for this study were collected in 2017 as part of an ongoing program impact assessment for a VSLG intervention in Mozambique. The study sample was drawn from women who were VSLG members and non-members residing in three sub-villages of the Manga Laforte area in the Sofala province, with a population of 64,626 [45,46]. The study employed a post-test-only quasi-experimental design, with VSLG members serving as the treatment group and non-VSLG members as the comparison group. A multi-stage selection method was used for sample selection, resulting in a total of 205 women included in the study. The study utilized publicly available data obtained from https://www.ncbi.nlm.nih.gov/pmc/articles/PMC6534299/, accessed on 12 March 2020. The original data collection for evaluating the VSLG program adhered to ethical protocols, including informed consent from participants. Approval for the secondary analysis of de-identified data was granted by the Institutional Review Board (IRB ID # 201706782) at the University of Northern Iowa.

For the treatment group, 19 VSLGs were randomly selected from a pool of 31 VSLGs operating in the three sub-villages of Manga Laforte. Subsequently, four to six study respondents were randomly chosen from each of the 19 selected VSLGs, resulting in a total of 105 women VSLG study respondents. To form the comparison group, community household registries were utilized to identify non-VSLG participants from the same three sub-villages. From these records, 100 non-VSLG participant women were randomly sampled, with approximately 30–35 female respondents selected from each sub-village. It is noteworthy that almost all the selected respondents (205 participants) in both the treatment and comparison groups were married. The demographic characteristics of the participants reflect the composition of the study sample and provide contextual information regarding the marital status of the participants.

#### 2.1.1. Dependent Variables

The outcome variable for this study is the availability of food in the household, measured using a shortened version of the household hunger score (HHS). The HHS scale has been validated and widely used across various cross-cultural settings in SSA and the Middle East, including Mozambique, South Africa, Kenya, Zimbabwe, and Gaza [61,62]. Other studies have also employed the same scale to measure household food security, referring to it as the 3 Item 3 Frequency (3I3F) scale [63,64] (Deitchler et al., 2011, 2010). Both the HHS and 3I3F scales utilize three Likert-type questions. The difference between these two tools lies in the time range that individuals are asked to recall regarding their household’s food availability patterns. The HHS assesses the pattern of food availability over the past 30 days, while the 3I3F scale only evaluates consumption frequency during the previous three days. The three typical questions included in the scale are as follows: (1) How often did you go a whole day and night without eating? (2) How often did you or someone in your household sleep hungry at night? (3) How often did you not have food to eat of any kind in your household? Participants are asked to indicate the frequency of these occurrences using four response options: never (0), rarely (1), sometimes (1), and often (2). The total score ranges from 0 to 6, with a cut-off point of 0–1 indicating “little to no hunger”, 2–3 indicating “moderate hunger”, and 4–6 indicating “severe hunger” [62,63,65]. The internal consistency of the measurement tool varied across different geographical settings, with a strong Cronbach’s α value of up to 0.93 reported [61] (Jesson et al., 2021). For this study, a 3-item questionnaire was utilized, and female respondents were asked to report their feelings over the past three days. A Cronbach’s α of 0.86 was reported for the scale.

#### 2.1.2. Independent and Mediating Variables

The independent focal variable for this study is participation in VSLGs, and the hypothesized mediator is household assets. Participation in a VSLG was measured using binary responses of “Yes” (coded 1, indicating participation in VSLG) and “No” (coded 0, indicating non-participation). Ownership of household assets was measured with a 20-item questionnaire where participants were asked to report whether or not their household had possessions such as a radio, lantern, bicycle, television, iron, motorbike, refrigerator, land, cows, and other animals. The number of times different kinds of assets were owned was summed, creating an asset ownership variable ranging from 0 to 20 to measure ownership of household assets. In this study, demographic variables of household size, age, education, and income were controlled.

### 2.2. Analysis Workflow

To address the hypothesized relationship, a descriptive analysis using *t*-tests and structural equation modeling (SEM) with Stata version 18 was utilized. First, the measurement part of the model was assessed to ensure a good model fit. Next, the hypothesized structural relationships were examined.

## 3. Results

Table 1 presents the characteristics of the sample, which consisted entirely of 205 women. Among the sample, 48% (100) were non-VSLG members (comparison group), and 51.2% (105) were VSLG participants (intervention group). The Household Hunger Score (HHS), based on a sample size of 205 respondents, ranged from 0 to 6, with a mean of 2. (SD = 1.97). Within this sample, 36% (73) scored little to no hunger (0–1), 33% (66) experienced moderate hunger (2–3), and 31% (63) faced severe hunger (4–6). In relation to ownership of household assets, the value range was between 0 and 9, with a mean of 4.26 (SD = 2.01). The average monthly household income ranged from 150 MZN to 50,000 MZN (1 $ = 50 MZN), with a mean of 3885.96 MZN (SD = 4335.76). The number of family members ranged from 1 to 10, with a mean of 5.4 (SD = 1.7). The ages of the participants ranged from 17 to 65, with a mean of 33.4 (SD = 11.9). In terms of education, the sample was divided into two groups: high school (46.8% (96)) and less than high school (51.7% (106)).

Table 2 describes the variance in the mean score of participation in VSLGs by household hunger score, household assets, number of household members, and average household income, between the intervention (VSLG) and the study (non-VSLG) groups. The intervention group had a significantly lower hunger score of 2.05 (SD = 1.90, *p* < 0.001) than the total sample, 2.52 (SD = 1.97), and the comparison group, 3.00 (SD = 1.95). The VSLGs also showed a significantly higher mean of ownership of household assets, 4.60 (SD = 2.10, *p* < 0.005), than the non-VSLGs, 3.19 (SD = 1.87). The study groups also showed a significantly higher mean (4656.44, SD = 5401.03, *p* < 0.001) of household average monthly income compared to the comparison groups’ mean score of 3,033.52 (SD = 2472.86). Income was measured with Mozambican currency in Metical/MZN ($1 = 50 MTZ). The respondents in both groups showed a similar number of household members, age, and educational level (i.e., years in school).

Preliminary analysis of the bivariate relationships revealed that HHS was significantly and inversely correlated with VSLGs (r = −0.236, *p* = 0.001) and household assets (r = −0.332, *p* < 0.001). This suggests that VSLG participants had lower HHS scores, indicating reduced levels of hunger, and higher levels of household assets, which were associated with a lower probability of experiencing food insecurity among VSLG participants compared to non-VSLG participants. The SEM model had good model fit indices, with a non-significant chi-squared value (chi2 = 0.67) and Pclose (0.830). The model fit was acceptable, with both the comparative fit index (CFI) and the Tucker–Lewis index (TLI) showing perfect scores of 1.000 and 1.013, respectively, meaning the model matches the data very well. Additionally, the standardized root mean square residual (SRMSR) was 0.082, and the root mean squared error of approximation (RMSEA) was 0.000, both very low, further confirming that the model accurately represents the data. Per the SEM model in Figure 2, participation in VSLGs was a significant negative predictor of household hunger (β = −0.21, *p* < 0.01), meaning participation in VSLGs is associated with a lower household hunger score. The results also showed a significant positive association of participation in VSLGs with increased ownership of household assets (β = 0.15, *p* < 0.025), and, in turn, a significant inverse correlation of the ownership of assets with household hunger score (β = −0.30, *p* < 0.01). The Sobel test confirmed that, besides the direct association, participation in VSLGs had a significant indirect role in lowering household hunger through the ownership of household assets (Sobel indirect effect [β = −0.06, *p* = 0.05]). The results from the factor loadings of the household hunger scale (HHS) with three items show robust values (0.84, 0.83, and 0.79), indicating strong associations between the items and the latent factor of hunger. These values suggest that the items are highly correlated with the underlying construct of hunger, with loadings well above the commonly accepted threshold of 0.5–0.6, indicating a valid and reliable scale.

## 4. Discussion

Existing evidence suggest an association between financial inclusion on the one hand and development on the other in terms of productivity, growth, and living standards. Financial inclusion opportunities, such as VSLG community interventions, foster multiple life outcomes through financial skills and by allowing access to financial services and products [29,32]. Individuals’ accumulated savings facilitate micro-investments that allow asset and wealth building over time and improve food security [16]. Initiatives that foster access to affordable and beneficial financial services and products are vital for advancing economic well-being and food security, thereby improving the quality of life. The study examined the direct and indirect role of participation in VSLGs on household hunger (i.e., nonavailability of food in the household). The mediation effect was examined through women’s ownership of household assets. The results supported existing assumptions and provided insights into the effectiveness of community-based financial interventions in improving food security through positive change in the ownership of household assets among low-income women groups.

The bivariate statistics suggested that, compared to the general sample and the non-VSLGs sample, the intervention group (VSLGs) showed a significantly lower mean score in household hunger and a higher positive mean score in household assets and income. Similarly, the results from the SEM indicated significant direct and indirect (i.e., through ownership of household assets) roles of participation in a microloan and saving community intervention in lowering household hunger. Consistent with previous studies, the findings supported the notion that community-based financial interventions, including VSLGs, have the potential to improve financial and non-financial outcomes related to food security.

These findings are consistent with the existing empirical and theoretical evidence that postulates the advantage of the financial inclusion approach in addressing poverty and its complex effects that hinder economic and health stability. [25,26,32]. For example, globally, the World Bank promoted financial inclusion approaches targeting economic gender disparity and ensuring food security [23,24]. The livelihoods of rural households in Africa, particularly in Mozambique, largely rely on subsistence agriculture and small-scale income-generating activities, both of which are highly susceptible to seasonal climate fluctuations [6,32]. Despite Mozambique’s rich natural resources, the country remains one of the poorest in SSA. The country’s focus on industrial economic policy and commodity-based export production has failed to instigate the structural changes needed to create employment opportunities and significantly reduce poverty [66].

Like its SSA counterparts, approximately 46% of Mozambique’s population lives below the poverty line, with an estimated annual income of $1228 per capita [10,12,67]. The population contends with numerous challenges, including poor governance, economic hardships, and limited access to quality education and healthcare. Social issues such as high child and maternal mortality rates, school dropouts, unemployment, heightened vulnerability to HIV, and migration further exacerbate these challenges. In regions like Mozambique, where access to formal banking services is limited, this model allows individuals to save and borrow money, share interest rates and dividends, and invest in small-scale income-generating activities that can lead to improved mental health outcomes. VSLGs also play a role in enhancing coping skills and the uptake of mental health services, including diagnostic treatments for depressive disorders [53,68].

The VSLG strategy is an innovative community approach used to advance women’s wealth and empowerment through financial and interpersonal outcomes that improve the overall quality of life [27]. Both the financial and non-financial benefits of participation in VSLGs have a positive impact on resilience [29,31,33,34]. These findings support the use of VSLGs as an affordable and accessible community-initiated strategy in addressing household food insecurity among rural low-income communities, particularly among women. The VSLG community intervention could facilitate a sustainable solution to address factors that elevate poverty and exacerbate unstable living conditions, particularly among low-income populations in Mozambique and other SSA communities. Women commonly take more responsibility related to the condition of living in the household, including relating to food availability, relationships, and other culturally expected norms and values [45]. The advantage of inclusion in a VSLG in a rural setup where formal financial infrastructures are limited and demanding could enable women to not only accumulate savings but also enhance their agency through interaction and learning to make better choices as they share experiences and practice reciprocity with group members [2].

## 5. Limitations

Despite highlighting the valuable contributions of this study in examining an innovative self-help community approach to address household food availability issues among low-income women, certain limitations should be acknowledged. Further, because group assignment was not randomized, unmeasured confounding variables may have influenced the outcomes, and, as a result, the study’s ability to infer causal relationships is constrained. The study utilized cross-sectional data (i.e., no baseline data for reference and measuring the changes more accurately), which restricts the ability to establish a causal relationship between VSLG strategies and food security. There is a possibility of recall bias when respondents were asked to recall their household hunger experiences over the past thirty days. The accuracy and consistency of recall may vary among participants, potentially affecting the reliability of the reported data. The relatively small sample size (i.e., 205) also has a limitation as it limited the ability to test multiple interactions and moderations without risking low statistical power.

## 6. Conclusions

Despite the limitations, the findings of this study have substantial implications and contribute to implementing community-owned and sustainable interventions aimed at improving food availability among low-income agricultural communities in SSA. These findings highlight the need for further research with stronger study designs to deepen our understanding of the causal relationships between VSLG participation, household assets, and food security outcomes.

Importantly, this study provides valuable evidence from Mozambique, shedding light on the effectiveness of community-owned financial inclusion strategies in rural SSA contexts. It addresses a significant knowledge gap by examining the contribution of VSLGs in predicting food security outcomes among rural low-income women in Mozambique, considering their specific socioeconomic and demographic conditions.

Moving forward, it would be crucial to conduct more comprehensive studies to explore the potential of the VSLG approach in urban low-income communities, encompassing both homogeneous and heterogeneous populations. Such studies would help to broaden our understanding of the applicability and effectiveness of community-owned financial inclusion strategies across diverse contexts, ultimately promoting sustainable development and improved food security. 

## Figures and Tables

**Figure 1 foods-14-00212-f001:**
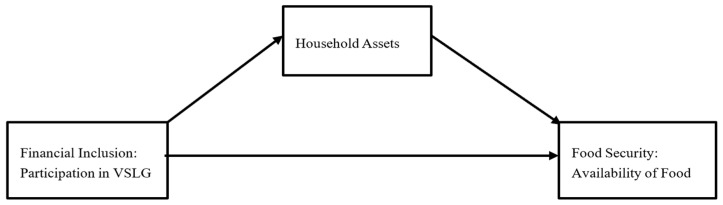
Conceptual framework illustrating the relationships between participation in VSLG, household assets (HAs), and food availability (HHS) among women in Mozambique.

**Figure 2 foods-14-00212-f002:**
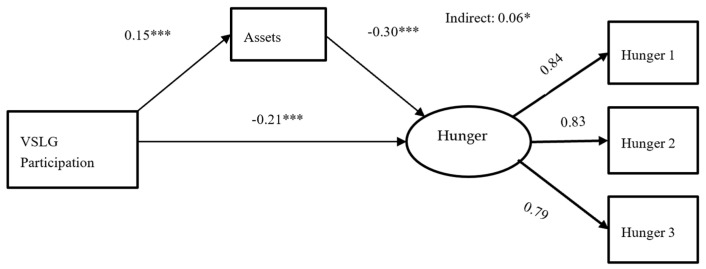
SEM complete mediation. Significance level: * *p* < 0.01; *** *p* < 0.001.

**Table 1 foods-14-00212-t001:** Descriptive statistics of variables linked with the household hunger score (HHS) in this study.

Variable	n (%) or Mean (SD)
Membership in a VSLG (N = 205)	
Yes	105 (48.8%)
No	100 (51.2%)
Household Hunger Score (N = 202)	
Mean (SD)	2.51 (1.97)
Median [Min, Max]	3.00 [0, 6.00]
Household Hunger Score Groups	
Little to no hunger	36% (73)
Medium hunger	33% (66)
Severe hunger	31% (63)
Household Assets	
Mean (SD)	4.26 (2.01)
Median [Min, Max]	4.00 [0, 9.00]
Average Family Income	
Mean (SD)	3890 (4340)
Median [Min, Max]	3000 [150, 50,000]
Family size	
Mean (SD)	5.37 (1.66)
Median [Min, Max]	5.00 [1.00, 10.0]
Age	
Mean (SD)	33.4 (11.9)
Median [Min, Max]	30.0 [17.0, 65.0]
Education Level	
High school	96 (46.8%)
Less than high school	106 (51.7%)

SD: standard deviation; Min: minimum; Max: maximum.

**Table 2 foods-14-00212-t002:** Sample characteristics by VSLG participation.

Variables	Frequency (%) or Mean (SD)		
	Total Sample (N = 250)	VSLG-Participants (n = 105)	Non-VSLG Participants (n = 100)
Household Hunger ***	2.52 (1.97)	2.05 (1.90)	3.00 (1.95)
Household Assets ***	4.26 (3.01)	4.60 (2.10)	3.19 (1.87)
Number of Family Members	1.48 (0.50)	1.53 (0.50)	1.49 (0.50)
Age	33.43 (11.88)	33.94 (11.55)	32.91 (12.26)
Education	2.47 (0.95)	2.57 (0.91)	2.38 (0.98)
Household Income (MZN; $1 = 50 MZN) ***	3885.96 (4335.76)	4656.44 (5401.03)	3033.52 (2472.86)

Frequency and mean are reported in the table; percentage and standard deviation are reported in parentheses. Significance level: *p* < 0.01; *** *p* < 0.001.

## Data Availability

The de-identified data for the study can be accessed and downloaded from the following link: https://www.ncbi.nlm.nih.gov/pmc/articles/PMC6534299/, accessed on 12 March 2020.

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
