# Peer review of "The Links Between Community-Based Financial Inclusion and Household Food Availability: Evidence from Mozambique"

_foods, 2025, doi:10.3390/foods14020212_

Round 1

Reviewer 1 Report (Previous Reviewer 2)

Comments and Suggestions for Authors

The authors have improved the manuscript and could be accepted in present form.

Author Response

Thank you so much for your valuable comment.

Reviewer 2 Report (New Reviewer)

Comments and Suggestions for Authors

1. This article explores “the link between community-based financial inclusion and household food supply.”

2. Upon examination of its content, the study provides policymakers and development experts with a potential strategy to leverage community-based financial inclusion approaches to improve the livelihoods of rural and low-income women.

3. This research also mentioned improving food supply and reducing hunger through asset building for low-income women in Mozambique.

4. The content of this manuscript is very different from the scope of the Foods journal.

5. Overall, this research is at the social science level and is more suitable for submission to SSCI journals. 

Author Response

Reviewer 3 Report (New Reviewer)

Comments and Suggestions for Authors

Introduction

The paper submitted for review analyzes the impact of participation in Village Savings and Loan Groups (VSLG) on household food security among low-income women in Mozambique. The study employs a quasi-experimental approach and structural equation modeling (SEM). The results indicate that participation in VSLG is associated with higher asset ownership levels and reduced household hunger. The study suggests that community-based financial inclusion strategies can effectively improve food security among women in rural areas.

Comments

The topic is significant in the context of global efforts toward sustainable development and the fight against poverty and hunger. The authors address an important socio-economic issue affecting a large proportion of Mozambique's population and other Sub-Saharan African countries.

The article provides robust evidence on the effectiveness of financial inclusion strategies in improving quality of life. However, it lacks a detailed discussion of potential cultural and regional differences that may affect the model's scalability.

While the authors mention previous studies, the literature review is limited and does not present a comprehensive overview of prior research on VSLG and their impacts on various aspects of household life in developing countries. It would be beneficial to elaborate on why previous studies were insufficient or how the new data and methodology used in this article complement existing knowledge. Including research from other Sub-Saharan African countries could also help contextualize the findings within a broader regional framework and assess their generalizability. Furthermore, the review should consider literature evaluating the effectiveness of similar microfinance programs to better compare their impacts on various aspects of household life. Expanding the literature review to include these aspects would strengthen the article’s argumentation and help clearly identify how the study addresses existing knowledge gaps.

The hypotheses are clearly stated, and the methodology employed (SEM) is appropriate for analyzing causal relationships. The authors utilized a control group consisting of women not participating in VSLG, which allowed for a comparison of outcomes between the groups. This approach increases the reliability of the results in the context of the research question. SEM is an advanced statistical method that enables the simultaneous examination of direct and indirect relationships between variables. In this case, it allowed for the investigation of how household assets mediate the relationship between VSLG participation and hunger reduction.

Despite these positive aspects, some limitations of the study should be addressed. The first limitation is the lack of randomized assignment to groups (VSLG participants vs. control group). This constrains the ability to draw definitive causal conclusions. There is a risk that participation in VSLG was correlated with participants’ characteristics (e.g., higher motivation, social resources), which may have influenced the outcomes independently of the intervention itself. The authors should discuss how they controlled for potential confounding variables (e.g., socio-economic status, education level, access to other forms of financial support).

Another limitation is the absence of a pre-post perspective, i.e., information on the baseline conditions of households before joining VSLG. Comparable data should also apply to the control group. The study was conducted using a post-test-only model, which prevents a comprehensive assessment of intervention outcomes. If the authors are unable to access baseline data (e.g., due to data unavailability), this limitation should be explicitly acknowledged. Additionally, the authors did not provide sufficient information on the demographic and economic similarities between the control group and VSLG participants. Differences in these aspects may affect the results and skew their interpretation. If the authors have data to characterize these two groups, this should be included in the text. If not, this limitation should be clearly stated.

Conclusion

In summary, the study's methodology is robust and suitable for the complex field conditions, but there are limitations that may influence the interpretation of the results. Addressing these shortcomings would significantly strengthen the conclusions drawn from the analyses. I therefore recommend that, following the incorporation of the suggested revisions

Author Response

Reviewer 4 Report (New Reviewer)

Comments and Suggestions for Authors

Overall, this is a good paper. However, there are a number of small issues, most of which require attention and that would increase its value, as detailed in the following. It is recommended that the authors give these serious consideration.

1.     Line 37: I believe there is data on food insecurity since 2021 (of course dramatically impacted by Covid) so maybe best to cite the latest numbers?

2.     Line 41: Many readers will know what SSA stands for but some will not. You first define it in line 147, but it would be best to do this in line 41.

3.     Line 44: You cite the years between 1900 and 2001 (i.e. one hundred years) whereas in line 42 you talk about the last three decades. Whether or not this is a mistake, population sizes in SSA have risen dramatically over both periods, so it’s not necessarily surprising that the absolute numbers experiencing food insecurity or showing malnourished individuals have also risen. It would be much better to state the actual rates as a percentage of the population at any particular time so the reader can better ascertain whether the situation has got worse, stayed the same or maybe even improved. Incidentally, evidence for the latter, i.e. that the situation has improved, is available prior to Covid.

4.     Section 1.1. Very useful section. Just a thought, is perhaps also to briefly reference other methods currently being deployed in SSA as elsewhere with some success in increasing financial inclusion, especially of women. For example, Muhammad Yunus’ famous microcredits have been well publicised for almost fifty years, and more recently direct giving (arguably a type of UBI but carefully targeted) seems to be having beneficial impacts (e.g. the GiveDirectly organisation). It may be too much to make comparisons with these, as well as other, methods, as this is not the purpose of this paper but it needs doing at some point – maybe this has been done?  The only independent evaluation of GiveDirectly I can find was published by the World Bank in 2013, so arguably needs an update: https://blogs.worldbank.org/en/impactevaluations/some-thoughts-give-directly-impact-evaluation. This blog also refers to the limitations of self-reporting and of power, the latter seemingly referring to the power of the conclusions derived from their data. The sample size issue is also addressed which is not done directly in your study.

5.     Line 121: ref 34 is difficult to find. Is this a UN publication? Please give the full reference, plus link, if possible.

6.     Lines: 144-146: being “less bureaucratic and demanding, as it primarily relies on trust, reciprocity, and meeting everyday needs, rather than extensive documentation such as proof of residency or income records” directly mirrors the documented advantages of microcredits, GiveDirectly, etc. Hence, at some point, there needs to be a detailed analysis and comparison between all these and related methods.

7.     Line 180: A sentence here that briefly defines social causation theory would be useful.

8.     Line 209: Figure 1 is very useful (as is Figure 2 later). Maybe later in the discussion it might be useful to briefly note that the three variables, once in operation will likely reinforce each other so that the three arrows would arguably each go in both directions. Also, the role of the other putative independent variables in Table 1 (average family income, family size, age and education level) might be anticipated in such a conceptual framework diagram. It is appreciated this is probably something for another paper – which I look forward to . See also comment 16.

9.     Lines 230-232: Table 1 shows that “yes” (the treatment group) consists of 100 respondents whereas line 230 gives the number of VSLG study representatives as 105. The reverse is the case for the comparison (“no”) group. Please check and correct if necessary.

10.  Line 242: Gaza is not in SSA.

11.  Lines 258-259: Doesn’t the HHS questionnaire ask about to last 30 days (see line 247) rather than three days? Although you state here a “3-item HHS questionnaire” so perhaps this is not the same as the HHS questionnaire mentioned in line 247? Please clarify and make the text unambiguous.

12.  Lines 271-274: For the intelligent lay-reader, it would be useful to briefly explain the SEM test, as well as “the RMSEA, CFI, and TLI”.

13.  Line 279: Is this 202 sample taken from the total 205 respondents? Is so, why only 202? Was the questionnaire not applied to all respondents? What is the spilt between “yes” and “no”. Similar questions apply to line 281 which mentions 198 respondents.

14.  Lines 293-302: The font size is too small, replicating the font of the note at the bottom of Table 1 as opposed to the main text font size.

15.  Lines 300-302: Please briefly explain the relevance and implication of the last sentence in the paragraph in intelligent lay-person terms.

16.  End of section 3: What I miss is any comment or analysis of the role of the other putative independent variables in Table 1 (average family income, family size, age and education level), given that data on these is presented. As in comment 8 above, I appreciate that this is probably something for another paper – which I look forward to – but maybe this should be stated in the conclusion alongside the other future work mentioned there .

17.  Lines 318-319: This sentence was not 100% clear on first reading. Maybe better to write: “Scholars indicated an association between financial inclusion, on the one hand, and development on the other in terms of productivity, growth, and living standards.”

18.  Line 321: There seems to be a reference number missing.

19.  Lines 361-362: These references should be numbered and placed in the reference list.

20.  Line 383-384: It is not clear what “cross-sectional data” means here? Perhaps it refers to the fact that your analysis doesn’t take account of the other independent variables in Table 1 (average family income, family size, age and education level) but lumps all these together? Please elucidate and also explain what a “randomized experimental design” means in intelligent lay-person terms. The paper should be accessible to non-statistical specialists.

21.  Lines 388-389: Yes, this is a general potential weakness of all quali-quantitative data collection approaches based on the use of the Likert scale, although all methods have strengths and weakness, so perhaps multi-method approaches might also be considered in future. See also comment 4 above for other possible limitations.

22.  Section 6: Overall, a very good conclusion but which could be extended with more information on useful follow-up analyses, such as examining the other dependent variables in Table 1. It is also evident that the use of different types of financial inclusion (like VSLG) are also relevant for non-food areas, both in terms of reducing poverty and increasing self-sufficiency and financial independence, education, health, entrepreneurship, community cohesion and development, etc. The conclusions are also generally in line with other financial inclusion methods, as mentioned above.

Round 2

Reviewer 2 Report (New Reviewer)

Comments and Suggestions for Authors

The education level of the respondents in this study includes high school and Less than high school. Are there any respondents with a college degree or higher? If so, it is recommended to amend the relevant description.

In Line 337: *p<.01 *** p<.01  Is it correct?

Some of the probability (p vaule) of correlation analysis in the manuscript are in italics, and some are not in italics. They should be adjusted to be consistent.

In Figure 2, please add note or description for * and ***. In addition, what is the message expressed by 0.84, 0.83, and 0.79?

Author Response

Comment 1: [The education level of the respondents in this study includes high school and Less than high school. Are there any respondents with a college degree or higher? If so, it is recommended to amend the relevant description].

Response: 1 [Thank you for this observation.  Unfortunately, the respondent’s educational level is set and cannot be changed since the data have already been collected.  The outcomes are only less than high school or high school and above]. 

Comment 2: [In Line 337: *p<.01 *** p<.01  Is it correct?]

Response: 2 [Thank you for your valuable comment: We have removed and replaced “Frequency and mean are reported in the table; percentage and standard deviation are reported in parentheses. Significance level: * p<.05; ** p<.01; *** p<.001 ( lines 338-339)].

Comment 3: [Some of the probability (p value) of correlation analysis in the manuscript are in italics, and some are not in italics. They should be adjusted to be consistent].

Response: 3 [We made corrections and changed to italic format (lines 359, 360)].

Comment 4: [In Figure 2, please add note or description for * and ***. In addition, what is the message expressed by 0.84, 0.83, and 0.79?]

Response: 4 [Thank you for your important comments. Now we have added Significance level: * p<.05; ** p<.01; *** p<.001” (line 373). We have also added the following for the additional comment

“The results from the factor loadings of the Household Hunger Scale (HHS) with three items show robust values, (0.84, 0.83, and 0.79), indicating strong associations between the items and the latent factor of hunger. These values suggest that the items are highly correlated with the underlying construct of hunger, with loadings well above the commonly accepted threshold of 0.5-0.6, indicating good validity and reliability of the scale (line: 364-368)”.

Comment 1:[The education level of the respondents in this study includes high school and Less than high school. Are there any respondents with a college degree or higher? If so, it is recommended to amend the relevant description].

Response: 1 [Thank you for this observation.  Unfortunately, the respondent’s educational level is set and cannot be changed since the data have already been collected.  The outcomes are only less than high school or high school and above]. 

Comment 2: [In Line 337: *p<.01 *** p<.01  Is it correct?]

Response: 2 [Thank you for your valuable comment: We have removed and replaced “Frequency and mean are reported in the table; percentage and standard deviation are reported in parentheses. Significance level: * p<.05; ** p<.01; *** p<.001 ( lines 338-339)].

Comment 3: [Some of the probability (p value) of correlation analysis in the manuscript are in italics, and some are not in italics. They should be adjusted to be consistent].

Response: 3 [We made corrections and changed to italic format (lines 359, 360)].

Comment 4: [In Figure 2, please add note or description for * and ***. In addition, what is the message expressed by 0.84, 0.83, and 0.79?]

Response: 4 [Thank you for your important comments. Now we have added Significance level: * p<.05; ** p<.01; *** p<.001” (line 373). We have also added the following for the additional comment

“The results from the factor loadings of the Household Hunger Scale (HHS) with three items show robust values, (0.84, 0.83, and 0.79), indicating strong associations between the items and the latent factor of hunger. These values suggest that the items are highly correlated with the underlying construct of hunger, with loadings well above the commonly accepted threshold of 0.5-0.6, indicating good validity and reliability of the scale (line: 364-368)”.

Comment 1:[The education level of the respondents in this study includes high school and Less than high school. Are there any respondents with a college degree or higher? If so, it is recommended to amend the relevant description].

Response: 1 [Thank you for this observation.  Unfortunately, the respondent’s educational level is set and cannot be changed since the data have already been collected.  The outcomes are only less than high school or high school and above]. 

Comment 2: [In Line 337: *p<.01 *** p<.01  Is it correct?]

Response: 2 [Thank you for your valuable comment: We have removed and replaced “Frequency and mean are reported in the table; percentage and standard deviation are reported in parentheses. Significance level: * p<.05; ** p<.01; *** p<.001 ( lines 338-339)].

Comment 3: [Some of the probability (p value) of correlation analysis in the manuscript are in italics, and some are not in italics. They should be adjusted to be consistent].

Response: 3 [We made corrections and changed to italic format (lines 359, 360)].

Comment 4: [In Figure 2, please add note or description for * and ***. In addition, what is the message expressed by 0.84, 0.83, and 0.79?]

Response: 4 [Thank you for your important comments. Now we have added Significance level: * p<.05; ** p<.01; *** p<.001” (line 373). We have also added the following for the additional comment

“The results from the factor loadings of the Household Hunger Scale (HHS) with three items show robust values, (0.84, 0.83, and 0.79), indicating strong associations between the items and the latent factor of hunger. These values suggest that the items are highly correlated with the underlying construct of hunger, with loadings well above the commonly accepted threshold of 0.5-0.6, indicating good validity and reliability of the scale (line: 364-368)”.

Comment 1:[The education level of the respondents in this study includes high school and Less than high school. Are there any respondents with a college degree or higher? If so, it is recommended to amend the relevant description].

Response: 1 [Thank you for this observation.  Unfortunately, the respondent’s educational level is set and cannot be changed since the data have already been collected.  The outcomes are only less than high school or high school and above]. 

Comment 2: [In Line 337: *p<.01 *** p<.01  Is it correct?]

Response: 2 [Thank you for your valuable comment: We have removed and replaced “Frequency and mean are reported in the table; percentage and standard deviation are reported in parentheses. Significance level: * p<.05; ** p<.01; *** p<.001 ( lines 338-339)].

Comment 3: [Some of the probability (p value) of correlation analysis in the manuscript are in italics, and some are not in italics. They should be adjusted to be consistent].

Response: 3 [We made corrections and changed to italic format (lines 359, 360)].

Comment 4: [In Figure 2, please add note or description for * and ***. In addition, what is the message expressed by 0.84, 0.83, and 0.79?]

Response: 4 [Thank you for your important comments. Now we have added Significance level: * p<.05; ** p<.01; *** p<.001” (line 373). We have also added the following for the additional comment

“The results from the factor loadings of the Household Hunger Scale (HHS) with three items show robust values, (0.84, 0.83, and 0.79), indicating strong associations between the items and the latent factor of hunger. These values suggest that the items are highly correlated with the underlying construct of hunger, with loadings well above the commonly accepted threshold of 0.5-0.6, indicating good validity and reliability of the scale (line: 364-368)”.

This manuscript is a resubmission of an earlier submission. The following is a list of the peer review reports and author responses from that submission.

Round 1

Reviewer 1 Report

Comments and Suggestions for Authors

This is a quasi-experimental study that explored the effect of participation in Village Saving and Loan Groups on household food availability, asset owner ship, and IPV among low-income women in central Mozambique.

The manuscript is well written. The introduction, although long, makes it possible to understand the problem even for those who are unfamiliar with Mozambique.

The variables and analysis methods are well defined; and the results described appropriately.

The discussion is limited and could be more in-depth. Therefore, further discussion of the results is suggested.

Reviewer 2 Report

Comments and Suggestions for Authors

 The manuscript, titled “The Role of Financial Inclusion on Availability of Food: Evidence from Southern Africa” describes the effects of financial inclusion on food availability and intimate partner violence in perspective of Sub-Saharan Africa. The authors analyzed data from Mozambique to evaluate the effects of participation in village savings and loan groups on household food availability, asset ownership, and intimate partner violence (IPV) among low-income women. The study focuses on livelihood improvement approaches, including food availability. While the language quality is good, the manuscript lacks readability for the general readers of the journal Foods. The design of the study and the organization of the findings are very poor and confuging. The authors fail to clearly express their goals and the objectives of the work. Additionally, the study design and scope do not align well with the focus on food analysis and policy. Therefore, I recommend rejecting the manuscript for publication in Foods.

Some specific comments:

1.      Introduction is too long. You should address main features of the study in 4-5 paragraphs.

2.      Introduction: Page 1; Line 39: Food insecurity represents a global health concern, resulting in an estimated 300,000 deaths each year. Could you please add reference?

3.      Introduction: Page 1; Line 45: “Tragically, this inadequate food consumption has led to the death of 3.5 million children under the age of five in SSA” This statement also require reference.

4.      Introduction: Page 2; Line 72:  Delete the elaborative term “Sub-Saharan Africa” because it is used earlier (Line 40).

5.      Introduction: Page 2; Line 77:  Delete (Anderson & Silva, 2020; CIA, 2020; Gradín & Tarp, 2019).

6.      The references in the text need to modify throughout the manuscript.

7.      Results: This section should focus on findings rather statistical features presentation.

8.      Discussion: It should be improved adding the reasons of the findings and comparing with previous reports.

Comments on the Quality of English Language